# Learned morphological features guide cell type assignment of deconvolved spatial transcriptomics

**Eduard Chelebian**[1]                                                EDUARD.CHELEBIAN@IT.UU.SE

**Christophe Avenel**[1]                                              CHRISTOPHE.AVENEL@IT.UU.SE

**Julio Leon**[2]                                                            JULIO.LEON@UCSF.EDU

**Chung-Chau Hon**[3]                                                CHUNGCHAU.HON@RIKEN.JP

**Carolina Wählby**[1]                                                CAROLINA.WAHLBY@IT.UU.SE

[1] *Department of Information Technology and SciLifeLab, Uppsala University, Uppsala, Sweden*

[2] *Department of Neurology and Weill Institute for Neurosciences, University of California, San Francisco, San Francisco, California*

[3] *Laboratory for Genome Information Analysis, RIKEN IMS, Yokohama, Japan*

**Editors:** Accepted for publication at MIDL 2024

## Abstract

Spatial transcriptomics enables to study the relationship between gene expression and tissue organization. Despite many recent advancements, existing sequencing-based methods have a spatial resolution that limits identification of individual cells. To address this, several cell type deconvolution methods have been proposed to integrate spatial gene expression with single-cell and single-nucleus RNA sequencing, producing per spot cell typing. However, these methods often overlook the contribution of morphology, which means cell identities are randomly assigned to the nuclei within a spot. In this paper, we introduce MHAST, a morphology-guided hierarchical permutation-based framework which efficiently reassigns cell types in spatial transcriptomics. We validate our method on simulated data, synthetic data, and a use case on the broadly used Tangram cell type deconvolution method with Visium data. We show that deconvolution-based cell typing using morphological tissue features from self-supervised deep learning lead to a more accurate annotation of the cells.

**Keywords:** self-supervised learning, spatial transcriptomics, cell type deconvolution

## 1. Introduction

Spatial transcriptomics has advanced our ability to understand the interplay between gene expression and tissue morphology, i.e., the spatial organization of tissue (Bressan et al., 2023). However, these methods, broadly classified into imaging-based and sequencing-based, are not without their limitations. Imaging-based methods reach sub-cellular resolution, but have limited gene coverage, while sequencing-based approaches, like Visium HD from 10X Genomics, Stereo-seq (Xia et al., 2022) and Seq-scope (Cho et al., 2021), compromise spatial resolution, and each sequenced tissue region may contain multiple cell types. To address this, several studies have proposed integrating spatial transcriptomics with single cell and single nucleus RNA sequencing (sc/snRNA-seq) by developing cell type deconvolution methods (Chen et al., 2022; Li et al., 2022, 2023). These methods can be categorized into probabilistic-based, non-negative matrix factorization-based, graph-based, deep learning-based and optimal transport-based (Li et al., 2023).

Benchmark studies show that Tangram (Biancalani et al., 2021), a deep learning-based method, and Cell2location (Kleshchevnikov et al., 2022), a probabilistic-based method, consistently outperformed others on various metrics. Interestingly, despite both methods utilizing nuclei segmentation from hematoxylin and eosin (H&E) for estimating cell density, they overlook morphology as a guiding factor for cell typing. Instead, Tangram assigns a cell types to each detected nucleus randomly. Recent efforts, such as SpaDecon (Coleman et al., 2023), tried to address this limitation by incorporating histology intensity values per region. However, this approach falls short of leveraging the rich information available in morphology. Consequently, while deconvolution accuracy at the spot level may be achieved, arguably the random attribution of cell types to individual nuclei does not allow a real increase in resolution.

To address this issue, we conceptualize the assignment as a problem of permutation. We hypothesize that we know the number of cell types within each spot from the deconvolution method, but that we have a permuted version of the actual composition. Using nuclei morphology as a guide, we conducted efficient hierarchical permutations under the assumption that similar cell types exhibit comparable nuclear morphology in H&E staining. To capture morphology we tried both classical morphology features and self-supervised deep representations. Due to the intrinsic difficulty of evaluating the method without a ground-truth, we conducted experiments on simulated and synthetic data as well as on a real use-case.

The main contributions of our work can be summarized as follows:

1. We developed a morphology-based cell re-assignment step for single-cell to spatial transcriptomics deconvolution.

2. We propose a hierarchical permutation method that allows to efficiently improve the arrangement of cell types in a tissue.

3. We used self-supervised deep learning features as powerful representations of cells.

MHAST (Morphology-guided Hierarchical reAssignment of cell types in Spatial Transcriptomics) can be integrated into any deconvolution method to achieve their full potential by leveraging the tissue morphology. The code for implementations and demos is available at https://github.com/eduardchelebian/mhast.

## 2. Methods

### 2.1. Mathematical formulation

The proposed approach seeks to efficiently determine the optimal arrangement of cell types in spatial transcriptomics experiments by addressing the computational challenge associated with exhaustive permutation calculations (with factorial complexity $O(n!)$). Instead of directly computing every possible permutation, the optimization is conducted in two hierarchical steps: first locally at the spot-level, and then globally. This hierarchical strategy reduces the permutation space, mitigating the complexity of the problem. Figure 1 shows the intuition behind the method.

Sequencing-based spatial transcriptomics experiments are organized in spots that capture transcriptome-wide gene expression. Given $N$ cells with $L$ cell type labels and $K$ morphological features belonging to $M$ spots of different sizes, let $A \in \{0,1\}^{N \times M}$ be the

one-hot encoded matrix indicating the belonging of each cell to a spot, let $B \in \mathbb{R}^{N \times K}$ be the matrix indicating the cell features and let $X \in \{0, ..., L\}^{N \times M}$ be the matrix indicating the cell type of each cell. Additionally, let $P_m \in \{0, 1\}^{N \times N}$ be the permutation matrix for rearranging cell type labels within a spot $m$.

First, we apply the optimization at spot-level locally. We define $\mathcal{P}_m$ as the space of all acceptable permutation matrices $P_m$ for spot $m \in M$ based on constraints (1) restrict permutations to each spot and (2) ensure that two cells of the same type are not permuted:

$$\forall m, \quad P_m \cdot A = A \tag{1}$$

$$\forall m, \quad P_m \cdot X \neq X \tag{2}$$

For each spot $m$, exhaustively find $P_m \in \mathcal{P}_m$ that maximizes the within-spot Calinski-Harabasz (CH) score (Caliński and Harabasz, 1974) for spots that have more than one cell of each type. Note that the choice of the CH score is deliberate, as it effectively balances between-cluster and within-cluster distributions, contributing to the method's efficacy.

$$\max_{P_m \in \mathcal{P}_m} CH(P_m \cdot X_m, B_m) \quad \text{if} \quad \exists l \in L_m : |X_{m,l}| > 1 \tag{3}$$

where the CH score for $L$ number of cell types on the dataset $B = [b_1, b_2, ..., b_N]$ is:

$$CH(X, B) = \frac{\sum_{l=1}^{L} |X_l| \, \|c_l - c\|^2}{L - 1} \Big/ \frac{\sum_{l=1}^{L} \sum_{i=1}^{|X_l|} \|b_i - c_l\|^2}{N - L} \tag{4}$$

where $|X_l|$ is number of cells with the $l$th label, $c_l$ is the centroid of the $l$th label and $c$ is the global centroid.

This step results in the arrangement of cells within each spot. However, it does not optimize arrangements within spots where the number of cells for each cell type is identical $|X_{m,i}| = |X_{m,j}|$ for $i \neq j$. In the case where $|X_{m,i}| = |X_{m,j}| = 1$, it is not possible to calculate the CH score, while if $|X_{m,i}| = |X_{m,j}| > 1$ there are multiple $P_m$ that yield the same highest CH score.

With these $m$ spots we define a new space of acceptable permutations $\mathcal{P}'_m \subset \mathcal{P}_m$. Having significantly reduced the permutation space, we can now conduct a global optimization. For each spot $m$, exhaustively find $P_m \in \mathcal{P}'_m$ that maximizes the across-spot CH score:

$$\max_{P_m \in \mathcal{P}'_m} CH(P_m \cdot X, B) \tag{5}$$

which will produce the arrangement for the rest of the instances.

## 2.2. Cell type deconvolution

In order to get a single cell type per detected nucleus in sequencing-based spatial transcriptomics experiments, we need to first segment the cells and then run the deconvolution methods for inferring the cell type composition.

**Nuclei segmentation.** For segmenting the nuclei we used the built-in nuclei detection method in QuPath (Bankhead et al., 2017) on the H&E image. This method has a good balance between speed and accuracy, enabling the efficient annotation of nuclei within a specified region of interest. Additionally, it provides measurements associated with the

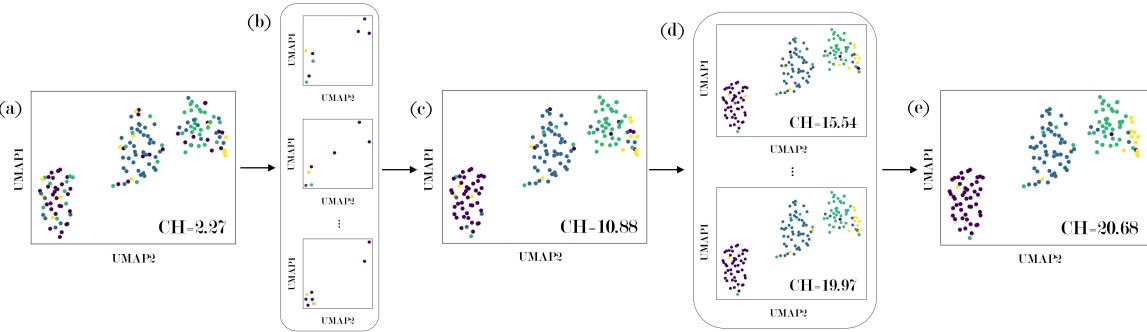

Figure 1: UMAP ([Becht et al., 2019](#)) dimensionality reduced feature space of (a) randomly permuted spots, (b) local optimization per spot, (c) result of local optimization, (d) global optimization and (f) final result. Values in the bottom right correspond to Calinski-Harabasz (CH) score. Colors represent different cell identities.

detected nuclei, serving as features for subsequent analyses, as presented in Section 2.3. Once the nuclei are detected, they can be used to measure the relative cell abundance in each spatial transcriptomic spot, which serves as a surrogate of cell density.

**Composition inference.** Following nuclei segmentation, the determination of cell type composition involves the application of deconvolution methods. As established before ([Chen et al., 2022](#); [Li et al., 2022](#), [2023](#)), Tangram ([Biancalani et al., 2021](#)) consistently emerges as one of the top-performing methods. Given its proven efficacy, Tangram was selected for validation in our experimental framework. Tangram is a deep learning-based approach which aligns single-cell gene expression data with spatial gene expression data by mapping them onto the same anatomical region, using shared genes for the mapping.

### 2.3. Feature extraction

From the detected nuclei, we extract morphological descriptors which will guide the permutations under the hypothesis that similar cell types share morphological features.

**Classical features.** Using QuPath ([Bankhead et al., 2017](#)), the same software utilized for detecting the nuclei, we extract per-nucleus descriptors and per-cell descriptors. Cells are defined by expanding each detected nucleus to a radius of 5 $\mu m$ until it encounters another nucleus expansion. This is not the most accurate estimation but serves as a way of including the context around the nucleus. The classical features extracted from the cells (C: cells) and nuclei (C: nuclei) include: area, perimeter, circularity maximum and minimum diameter, eccentricity and H&E-derived intensity features.

**Self-supervised learning features.** Our dataset was too small to expect the model to learn relevant features training it from scratch. We therefore started from a publicly available ResNet18 model trained by self-supervision with SimCLR ([Chen et al., 2020](#)) on 57 histopathological datasets ([Ciga et al., 2022](#)). We fine-tuned the model using the detected nuclei as centers, extracted one image patch per cell, and trained by self-supervision with SimCLR in the same way as in the original model. Experiments included patch sizes of $32 \times 32$ (DL: 32), $64 \times 64$ (DL: 64) and $128 \times 128$ (DL: 128) to test the contribution of

different contexts. For example patches, see Appendix F. Finally, we used the fine-tuned model's last fully connected layer before prediction to define features for each cell patch.

## 3. Experiments on simulated data

We first evaluated the method using simulated Visium data, enabling generation of ground-truth for methodological validation under controlled conditions.

### 3.1. Data generation

We are essentially simulating the relative abundance of cell types per spot, which is the output from deconvolution methods. This output is then randomly assigned to the detected nuclei and denoted as $X_{perm}$, as shown in Figure 2. Subsequently, guided by the morphological features in the feature matrix $B$, our objective is to optimally assign cell types to each morphology, resulting in the actual assignment denoted as $X$. The details of the data simulation can be found on Appendix A.

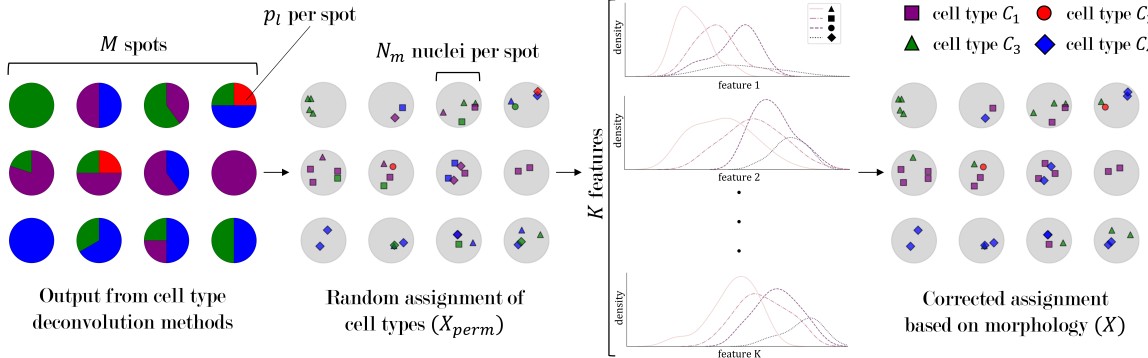

Figure 2: Data simulation workflow. From the output from cell type deconvolution methods, we simulate the random assignment of cell types ($X_{perm}$). Guided by the simulated morphological features $B$, we correct the cell assignment to match the cell identities with their morphology ($X$). More details on Appendix A.

### 3.2. Evaluation and results

The evaluation of the simulated data aims to determine the level of feature descriptiveness required to recover the true cell type arrangement $X$ from its permuted version $X_{perm}$.

Appendix B shows how the rearrangement accuracy of $X_{perm}$ with respect to the original $X$ changes when increasing the feature overlap. The baseline is established by calculating the accuracy of the randomly permuted $X_{perm}$. With non-overlapping features for each cell types, we have a perfect rearrangement accuracy. Notably, as feature overlap increases, the rearrangement still remains valuable, outperforming random allocation.

## 4. Experiments on synthetic data

One approach to incorporate actual H&E features into the evaluation, while still having a ground-truth, is to generate a synthetic Visium dataset from other spatial transcriptomics

methods. Xenium from 10X Genomics is particularly suitable for this purpose. It provides cell typing information from high-resolution imaging-based spatial transcriptomics, typically accompanied by DAPI imaging, but also includes H&E staining on the same section.

### 4.1. Dataset

We use the Xenium In Situ Breast Dataset[1], by 10X Genomics (Janesick et al., 2022). We sample a region with variable density of cells and cell types and generate synthetic Visium spots including the cell type locations. The details can be found in Appendix C.

### 4.2. Evaluation and results

The assessment of the synthetic data focuses on identifying the descriptors that yield the best reconstruction score. To this end, we implemented the method with the different feature extractors and calculated the macro-averaged F1-score to capture the overall contribution of all cell types. This evaluation is compared with random permutations, which emulate the result from deconvolution methods.

The results in Figure 3 reveal that applying the method consistently enhanced the results, outperforming the majority of random outcomes. In fact, self-supervised features on $64 \times 64$ patches surpass every random results and successfully retrieves more of the original cell types. It is important to note that this task is more challenging than its real word counterpart, given the higher density and diversity of cell types in comparison to what one would encounter in a typical Visium experiment.

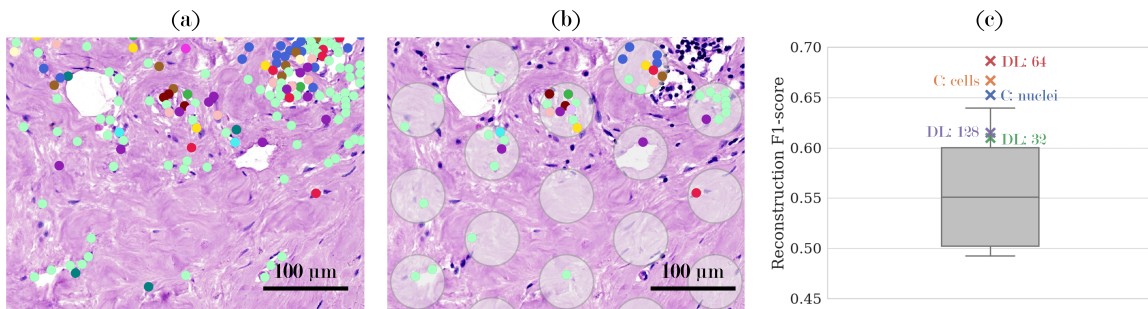

Figure 3: (a) Region of Xenium cell typing registered on H&E. (b) Region of synthetic Visium data. (c) Reconstruction F1-score boxplot from random permutations. DL: 32, DL: 64 and DL: 128 represent self-supervised features with patch sizes of 32, 64 and 128, respectively. C: nuclei and C: cells represent the classical features in the detected nuclei and on the expanded cells, respectively.

## 5. Experiments on real data

Having assessed the robustness of the hierarchical permutation method on simulated data, and confirmed that self-supervised features from $64 \times 64$ patches as the most effective within synthetic data, we applied our method to a real-world use case.

---

1. https://www.10xgenomics.com/products/xenium-in-situ/preview-dataset-human-breast

### 5.1. Dataset

To validate the method, we employed the same datasets from the original Tangram paper (Biancalani et al., 2021). This dataset is a 10X Genomics Visium experiment on a mouse brain coronal section[2]. For annotated scRNA-seq data, we used the mouse cortex dataset shared by (Tasic et al., 2018). Both the single-cell and spatial datasets are publicly available in Scanpy (Wolf et al., 2018) and Squidpy (Palla et al., 2022) APIs, ensuring reproducibility.

### 5.2. Evaluation and results

Using QuPath, we located the nuclei within the H&E image associated with the Visium mouse brain. Tangram was then applied to two regions (refer to Appendix D) within the cortex of the mouse brain, utilizing annotated scRNA-seq data for cell type deconvolution. We employ self-supervised learning to extract morphological features from $64 \times 64$ patches centered on the nuclei, and our method is then applied to rearrange the cell types based on these features. Since a ground-truth is not available, evaluating our approach in this context is not straightforward. Nevertheless, we can assess whether the global score attained through our two-step optimization surpasses the global score from random permutations. To accomplish this, we perform 10000 random shuffles within the spots and examine whether our method shows superior performance. This evaluation is conducted using both the CH score —the one we are trying to maximize—, and also the Davies-Bouldin score (Davies and Bouldin, 1979), which we did not try to minimize.

Figure 4 shows the results on region 1. The density plots in Figures 4c and 4d demonstrate that our two-step optimization indeed maximized the CH score and minimized the Davies-Bouldin score in comparison with the random rearrangements. This suggests that our method achieves a rearrangement that ensures consistency in morphology for each cell type. For the sake of reproducibility, we repeated the analysis on another region of the cortex, as detailed in Appendix E, obtaining similar results.

## 6. Discussion and Conclusions

In this paper, we presented MHAST, an efficient permutation method for rearranging the cell types from spatial transcriptomics-single cell deconvolution guided by self-supervised morphology features.

Using simulated data and morphological features, we demonstrated the effectiveness of the method in reconstructing the original arrangement of cell types guided by their morphology. We additionally established the method's robustness by progressively reducing the descriptive power of the features, yet still showing the value of applying the method.

Through the use of synthetic Visium data generated from Xenium, we incorporated real H&E morphological features alongside a form of ground truth. Despite the challenges associated with registering one image modality to another and synthesizing the data, the results indicated that self-supervised features outperformed other descriptors in characterizing morphology and reconstructing the original arrangement. The differences between patch sizes can be attributed to the amount of context, as explored in Appendix F.

---

2. https://support.10xgenomics.com/spatial-gene-expression/datasets/1.1.0/V1_Adult_Mouse_Brain

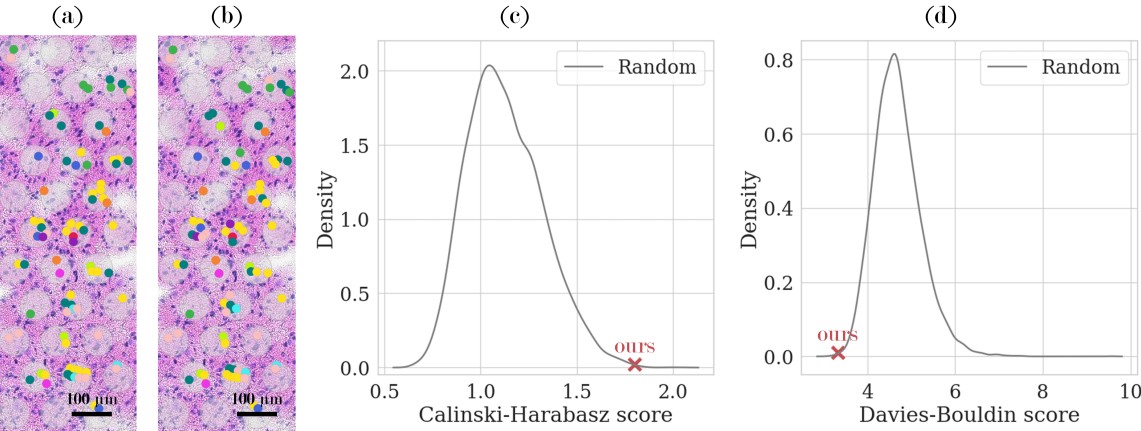

Figure 4: Results for region 1. (a) Tangram randomly assigned cell types. (b) Corrected cell types using MHAST. (c) Calinski-Harabasz (higher is better) and (d) Davies-Bouldin (lower is better) scores for 10000 random bag permutations and our method. Interactive visualization available in TissUUmaps (Pielawski et al., 2023) at https://mhast.serve.scilifelab.se/brain_mouse.tmap.

In the practical application of the method to Tangram as a use case, our experiment revealed that the two-step rearrangement achieved results equivalent to the best possible global arrangement. Importantly, this was achieved without incurring the computational costs associated with global permutations.

An inherent limitation of the method lies in the assumption that every cell type possesses an identifiable morphology that can be leveraged in this problem, which may not be applicable for every encountered cell type. For instance, non-neuronal cells like oligodendrocytes have an identifiable small and round nucleus surrounded by cytoplasm, while cells in different neuronal layers can be more challenging to distinguish in H&E. Another constraint arises when applying the method in large regions, where computational costs become prohibitive. This issue can be addressed by implementing the global optimization step in a rolling window manner.

Future lines of work may explore the prediction of cell types also outside the spots, based on the maximization of the permutation results. This could be done especially for cell types with a well-documented morphology. Another potential direction involves streamlining the process by incorporating the permutation step into the workflow of Tangram or other cell type deconvolution methods. This can be relevant as some of these methods already incorporate cell segmentation as part of their process.

In conclusion, MHAST is able to enhance the potential of cell type deconvolution methods such as Tangram by improving the attribution of cell types in low-resolution sequencing-based methods like Visium. The ideas of using self-supervised learning-based morphology and cluster tightness metrics to complement the information provided by molecular data could be extended further to applications beyond spatial transcriptomics.

## Acknowledgments

This research was funded by the European Research Council via ERC Consolidator grant CoG 682810 and Technology Development project from SciLifeLab to C.W. and support from the Scandinavia-Japan Sasakawa Foundation to E.C.

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

## Appendix A. Generating simulated data

For generating simulated Visium data, we need to define the number of spatial transcriptomics spots $M$, the range of cells per spot $N_m$, the number of cell types $L$ and the number of features $K$.

Given the number of spatial transcriptomics spots $M$ and a range of cells per spot $N_m$ we generate matrix $A \in \{0, 1\}^{N \times M}$ that indicates the belonging of each nucleus to a spot. Given the relative proportion of cell types $p_l$ for $l \in L$ we can define matrix $X \in \{0, ..., L\}^{N \times M}$ which indicates the belonging of each nucleus to a cell type (our ground truth). Finally, specifying the number of features $K$, we can define the features matrix $B \in \mathbb{R}^{N \times K}$. We work under the assumption that cell types have descriptive morphological features, thus we model this by sampling values from different distributions for each cell type. Features for each cell type $l$ are generated by adjusting the standard deviation based on an overlap $d$ sampled from $B_l \sim \mathcal{N}(l, d)$. This will further allow to compare also how different do the morphological features per cell type be for the method to work.

Finally, we apply random permutations per bag to matrix $X$ to generate $X_{perm}$. The synthetic problem then would be to find the permutation matrices $P_m$ per bag $m$ that allows to recover $X$ from $X_{perm}$.

Specifically, for the example in Figure 2 we used $M = 12$, $N_m = [2, 6]$, $L = 4$ and $K = 10$. We chose the proportion of cells $p_l$ such that the first row contains a higher percentage of cell type $C_3$, the second $C_1$ and the third $C_4$, with $C_2$ being only marginally present, simulating different layers.

To give an idea of the efficiency gain even on this simulated dataset, if one were to exhaustively calculate the global optimal arrangement from all the possible permutations within every spot it would take, from left to right and from top to bottom, $1 \cdot 2 \cdot 20 \cdot 12 \cdot 5 \cdot 12 \cdot 10 \cdot 1 \cdot 1 \cdot 3 \cdot 6 \cdot 6 = 31104000$ operations. Using our proposed hierarchical approach it takes $1 \cdot 2 \cdot 2 \cdot 2 \cdot 1 \cdot 2 \cdot 1 \cdot 1 \cdot 1 \cdot 1 \cdot 6 \cdot 2 = 192$ operations, due to many spots having one or two local optima that are then included in the reduced permutation space.

## Appendix B. Results of simulated data

Figure 5 shows the reconstruction F1-score from applying the method with increasingly overlapping, and thus, decreasingly descriptive features.

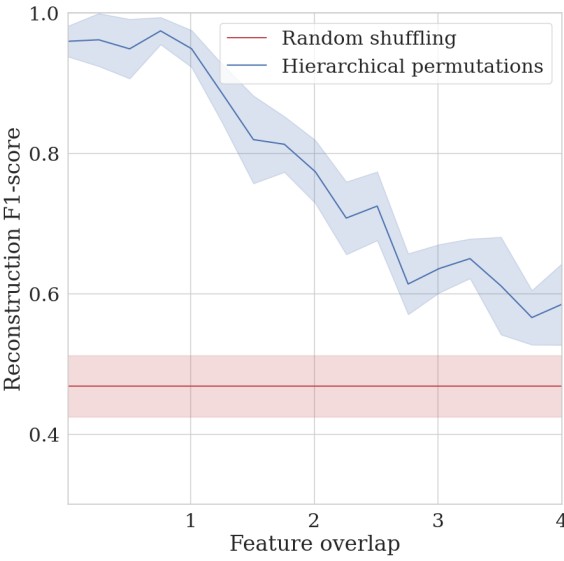

Figure 5: Reconstruction F1-score from random shuffling and applying the hierarchical permutation method with different levels of feature overlap. Shaded areas correspond to the 95% confidence interval from 10 initializations.

## Appendix C. Generating synthetic Visium data from Xenium

Xenium from 10X Genomics includes cell typing per cell along with H&E images. The first step involves cell typing on cells detected in another image stained with DAPI, requiring the registration of the DAPI image to the H&E image. Utilizing the scale-invariant feature transform (SIFT), we achieved this registration, although obtaining perfect cell-to-cell alignment between two different modalities proved challenging. Following H&E to DAPI image registration, QuPath was utilized to detect nuclei in the H&E images. Using the nuclei locations in H&E and the registered locations in DAPI, we mapped the closest cell types from the registered to the detected ones. This process inherently poses a challenge, as certain cell types may not be accurately mapped if another entity is closer, adding complexity to our subsequent task. With cell typing established on the H&E, the next step involves synthesizing Visium spots based on their distribution in real-life scenarios and discarding cells falling outside these spots.

The Xenium cell type annotations, DAPI image and H&E image used to generate the synthesize the Visium data are available at https://www.10xgenomics.com/products/xenium-in-situ/preview-dataset-human-breast.

## Appendix D. Extracted regions from full image

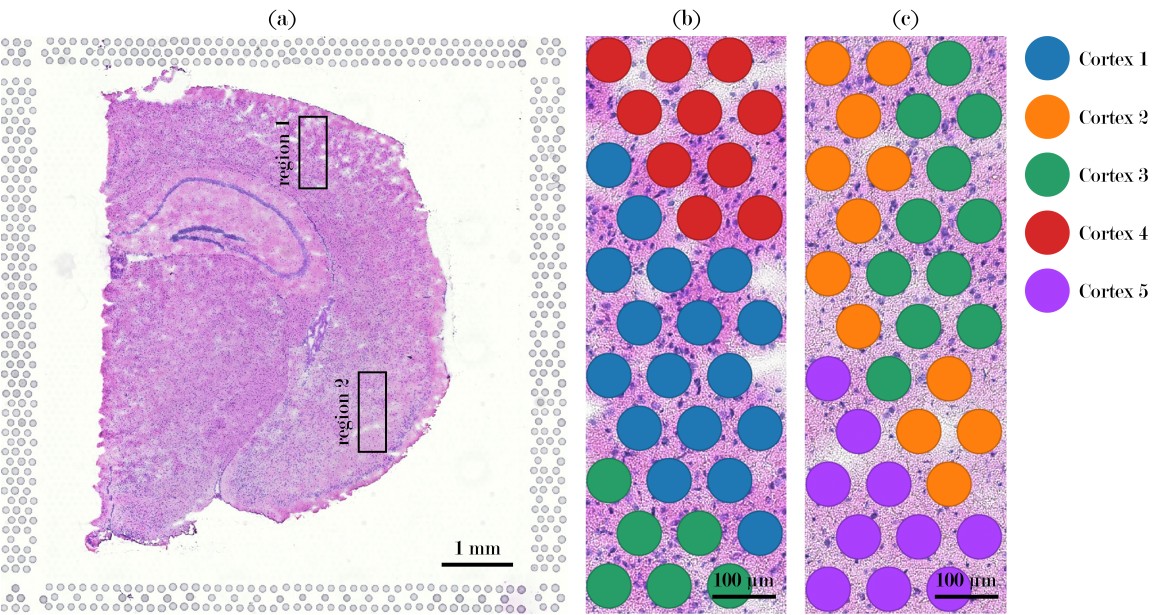

Figure 6: (a) Original Visium image with regions 1 and 2. (b) and (c) Regions 1 and 2 annotated spots. Interactive full resolution visualization available in TissUUmaps ([Pielawski et al., 2023](#)) at https://mhast.serve.scilifelab.se/brain_mouse.tmap.

## Appendix E. Analysis on region 2

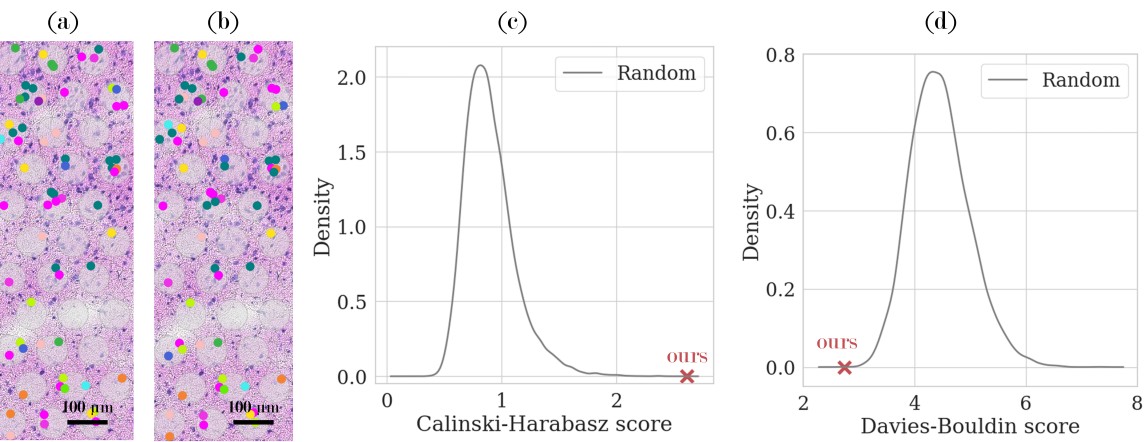

Figure 7: Results for region 2. (a) Tangram randomly assigned cell types. (b) Corrected cell types using MHAST. (c) Calinski-Harabasz (higher is better) and (d) Davies-Bouldin (lower is better) scores for 10000 random bag permutations and our method. Interactive full resolution visualization available in TissUUmaps (Pielawski et al., 2023) at https://mhast.serve.scilifelab.se/brain_mouse.tmap.

## Appendix F. Effect of patch size

Figure 8 shows the different patch sizes extracted with the same neighboring nuclei as centers. We hypothesize 32×32 patches do not include enough context for the self-supervised method to learn meaningful representations, while 128 × 128 patches include too much context and nuclei that are close start having similar features, hindering their separation per type.

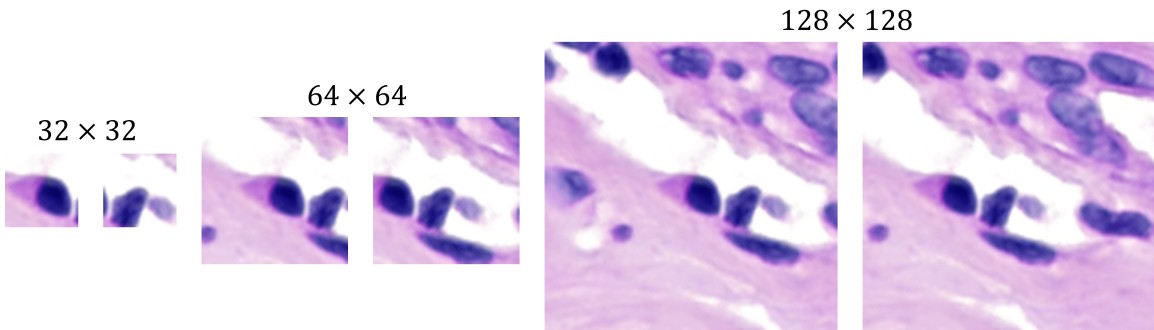

Figure 8: Comparison of patch sizes for two neighboring cells.

