# OpenReview forum: "Learned morphological features guide cell type assignment of deconvolved spatial transcriptomics"
_MIDL.io/2024/Conference — MIDL 2024 Poster_

### Official Review · Reviewer_W2yE · 2024-02-26

**Confidence:** 3
**Preliminary Rating:** 4
**Final Rating:** 4

**Summary:**

This paper introduces a novel hierarchical permutation-based framework that incorporates tissue morphology to guide cell type assignment in spatial transcriptomics, addressing the limitations of spatial resolution in existing sequencing-based methods. By integrating spatial gene expression with single-cell and single-nucleus RNA sequencing data, the proposed method aims to produce more accurate cell typing per spot. The method is validated on simulated, synthetic data, and a use case involving the Tangram cell type deconvolution method with Visium data, demonstrating that leveraging morphological features through self-supervised deep learning leads to annotations that more closely match true cell identities. This approach represents a significant advancement in the field by acknowledging and utilizing the often-overlooked contribution of morphology to cell identity.

**Strengths:**

- The paper tackles a critical challenge in spatial transcriptomics, offering a clear and concise contribution to the field.
- The innovative use of a hierarchical permutation-based framework to integrate morphological features into cell type deconvolution is both novel and impactful, potentially leading to significant improvements in the accuracy of cell type assignments.
- Making the source code publicly available greatly enhances the paper's value, facilitating reproducibility and further research in the community.
- The authors have comprehensively addressed future perspectives and limitations, demonstrating a well-rounded understanding and thorough investigation of their proposed method.

**Weaknesses:**

The term "more true" annotations is vague and lacks precision. It would benefit from clarification, possibly specifying if it refers to increases in true positives or overall accuracy.
A discussion on the proposed method in terms of computational efficiency and scalability compared to existing approaches, especially when applied to large spatial transcriptomics datasets is missing and would be appreciated.
What ist the potential applicability of the proposed method to other types of molecular or imaging data beyond spatial transcriptomics? The view on the authors on this point would be very interesting.
Finally I would appreciate the authors comment on the consideration of the method's robustness to variations in tissue morphology across different conditions or disease states, and how this might affect the generalizability of the approach.

**Detailed Comments:**

- Minor typo on page 2 ("levering" instead of "leveraging")
- Figure 4 would benefit from a clear indication of whether higher or lower values represent better scores. Adding this information to the figure's caption or directly on the plot would improve readability and comprehension of the results.

**Justification Of Final Rating:**

I want to thank the authors for the thorough revision of their work and for addressing all my raised questions and concerns.

In line with my initial assessment, I recommend to accept the revised research paper.

**Justification Of The Preliminary Rating:**

Overall the presentation of the proposed work is sound and well done. The contribution is clear and incorporates an interesting approach to incorporate tissue morphology to guide cell type assignment in spatial transcriptomics.

**Questions To Address In The Rebuttal:**

- Could the authors provide a more detailed explanation of what is meant by "more true" annotations? Specifically, does this refer to an increase in true positive rates, accuracy, or another metric?
- How does the proposed method compare to existing approaches in terms of computational efficiency and scalability, especially when applied to large spatial transcriptomics datasets?
- The paper would benefit from a discussion on the potential applicability of the proposed method to other types of molecular or imaging data beyond spatial transcriptomics. Can the authors comment on this aspect in their rebuttal?
- Is there any consideration of the method's robustness to variations in tissue morphology across different conditions or disease states, and how might this affect the generalizability of the approach?

---

> ### Author Response · Authors · 2024-03-14
> **Rebuttal for Reviewer W2yE (1/2)**
>
> &ensp; **1. Could the authors provide a more detailed explanation of what is meant by "more true" annotations? Specifically, does this refer to an increase in true positive rates, accuracy, or another metric?**
>
> We agree with the Reviewer that “more true” cell types is not a very precise term. At a high level, what we meant is that, after applying the method, we get both a higher cell type resolution and cell type identities which are closer to their true class. That is, if the deconvolution methods do not include the contribution of morphology, their cell typing can only be per-spot, so a random assignment would be further from what the tissue actually contains as compared to applying our method. In terms of a metric to express this, we chose to use the F1-score for its ability to balance between precision and recall. Thus, following the Reviewer’s comment, we consider that “more accurate” is a better way to frame it:
>
> > We show that deconvolution-based cell typing using morphological tissue features from self-supervised deep learning lead to a more accurate annotation of the cells.
>
> ---
>
> &ensp; **2. How does the proposed method compare to existing approaches in terms of computational efficiency and scalability, especially when applied to large spatial transcriptomics datasets?**
>
> As we specify in the Introduction there is, to our knowledge, one method —SpaDecon— that includes morphology for performing spatial transcriptomics deconvolution (Coleman et al., 2023). The morphology used by SpaDecon is inspired by another method (SpaGCN by Hu et al, 2021) and it uses means and variances of the RGB channels in $50\times50$ pixel patches. This approach significantly differs from ours in two ways: (1) the extracted features are arguably less descriptive than deep learning or even simple cell descriptors and (2) the features are extracted per patch and it results in a per-spot deconvolution instead of a per-cell deconvolution. We consider that these differences do not enable direct comparison of the results of SpaDecon with our Tangram followed by our permutation approach.
>
> Regarding the efficiency and scalability of our method, it is subject to the performance of the deconvolution method on which it is applied. For all experiments, the optimal permutation was found in a matter of minutes and, as we specify in the Discussion, the only constraint is the amount of unique cell types inside a given spot which can be prohibitive at extreme cases where the local step of our hierarchical permutation would yield too many local optima so the global step would still be too high. In these cases, we propose to apply the method in a non-overlapping sliding window manner which would ensure global optimum per window without incurring in the computational costs.
>
> Regarding the comparison between the proposed hierarchical permutation and a global permutation, we refer the Reviewer to Reviewer gGXC's question 6 and the updated Appendix A.
>
> ---
>
> &ensp; **3. The paper would benefit from a discussion on the potential applicability of the proposed method to other types of molecular or imaging data beyond spatial transcriptomics. Can the authors comment on this aspect in their rebuttal?**
>
> We appreciate the insight by the Reviewer. As it is clear from the paper, the use case for which we formulated this method was for sequencing-based spatial transcriptomics such as 10X Visium. We observed that while single-cell to spatial transcriptomics deconvolution methods were able to generate per-spot cell typing, if they wanted to get per-cell typing they had to assign the types randomly to the cells (Tangram by Biancalani et al. 2021, Figure 4a).
>
> We believe that the permutation framework itself would be mainly applicable to this type of data. However, the ideas of using self-supervised learning-based morphology and cluster tightness metrics (such as the Calinski-Harabasz score) to complement the information provided by molecular data could be extended further to applications beyond spatial transcriptomics. As we discussed in comment 2, we think that current methods fail to leverage all the morphological information available, so we hope these kinds of frameworks can contribute to utilize the full potential of morphology regardless of their specific data type. We added these insights to the discussion accordingly:
>
> > In conclusion, the proposed permutation method is able to enhance the potential of cell type deconvolution methods such as Tangram by improving the attribution of cell types in low-resolution sequencing-based methods like Visium. The ideas of using self-supervised learning-based morphology and cluster tightness metrics to complement the information provided by molecular data could be extended further to applications beyond spatial transcriptomics.

---

> ### Author Response · Authors · 2024-03-14
> **Rebuttal for Reviewer W2yE (2/2)**
>
> &ensp; **4. Is there any consideration of the method's robustness to variations in tissue morphology across different conditions or disease states, and how might this affect the generalizability of the approach?**
>
> As we specify in the updated section 2.3. Feature extraction thanks to Reviewer dLtb’s comments, we use a model pretrained on 57 histopathology datasets (Ciga et al., 2022) which includes tissue both from normal and various diseased conditions. That gives us the certainty that the model has been exposed to morphologies across different conditions already. Additionally, we perform fine-tuning using self-supervised contrastive learning on the dataset we analyze which ensures that, if there were any morphologies specific to the data we are analyzing, they would be represented in the feature space.
>
> ---
>
> &ensp; **5. Minor typo on page 2 ("levering" instead of "leveraging"). Figure 4 would benefit from a clear indication of whether higher or lower values represent better scores. Adding this information to the figure's caption or directly on the plot would improve readability and comprehension of the results.**
>
> We implemented both corrections.
>
> ---
>
> **References**
>
> Coleman, K., Hu, J., Schroeder, A., Lee, E. B., & Li, M. (2023). SpaDecon: cell-type deconvolution in spatial transcriptomics with semi-supervised learning. Communications Biology, 6(1), 378.
>
> Hu, J., Li, X., Coleman, K., Schroeder, A., Ma, N., Irwin, D. J., ... & Li, M. (2021). SpaGCN: Integrating gene expression, spatial location and histology to identify spatial domains and spatially variable genes by graph convolutional network. Nature methods, 18(11), 1342-1351.
>
> Biancalani, T., Scalia, G., Buffoni, L., Avasthi, R., Lu, Z., Sanger, A., ... & Regev, A. (2021). Deep learning and alignment of spatially resolved single-cell transcriptomes with Tangram. Nature methods, 18(11), 1352-1362.
>
> Ciga, O., Xu, T., & Martel, A. L. (2022). Self supervised contrastive learning for digital histopathology. Machine Learning with Applications, 7, 100198.

---

> > ### Comment · Reviewer_W2yE · 2024-03-21
> >
> > I want to thank the authors for thoroughly answering all my questions and concerns!
> > No further questions from my side.

---

### Official Review · Reviewer_dLtb · 2024-02-28

**Confidence:** 2
**Preliminary Rating:** 4
**Recommendation:** Poster
**Final Rating:** 4

**Summary:**

The authors argue that past work on cell type deconvolution does not make use of the morphology of the underlying tissue. To address that, they introduce a hierarchical permutation-based framework that leverages information from tissue morphology, extracted in the form of features from a self-supervised model, to reassign cell types. They perform evaluation of their method on simulated, synthetic and real data to show that cell type assignment improves with the proposed method.

**Strengths:**

- The paper is well written overall and easy to follow. Mathematical hypotheses are well justified and defined.
- The proposed work comes with a codebase to help reproducibility and transparency, which is a welcome contribution and addition.
- The appendices are actually helpful and a relevant addition to the main body of text.

**Weaknesses:**

- The method section on the deep learning application is not justified as well as the other sections. It may be considered enough as a proof of concept to show that extracting tissue morphology features does indeed help, but it could have been explored more. For example, other encoders could have been tested to this end.

**Detailed Comments:**

- I would maybe suggest to rethink about the order of subsections in the paper. Usually, all dataset subsections (simulated, synthetics, real) would be combined under the same section and be presented before any method section.

**Justification Of Final Rating:**

The authors addressed most of my questions during the rebuttal period, and I am mostly satisfied with their answers. However, my points were overall minor, so I will not change my initial rating. Moreover, my knowledge of the authors' field is limited, so I will be conservative by keeping the same rating.

**Justification Of The Preliminary Rating:**

I am overall satisfied with the content, presentation and structure of the paper. It is scientifically sound. I should however mention that my knowledge on spatial transcriptomics, genomics and cell type deconvolution is very limited. My evaluation is mostly based on the general methodology, mathematical development and the deep learning use for feature extraction. I am unsure about the impact of this work on the field of spatial transcriptomics.

**Questions To Address In The Rebuttal:**

- On the feature extraction with SimCLR (1) Have you thought about keeping the larger patch size (128), but extracting multiple hidden layers of features from the model (i.e. not only the last one)? That would maybe help obtain local (early layers) and global (last layer) patch features; (2) is there a justification on why SimCLR (2020) is used here? There are a plethora of recent self-supervised encoders and feature extractors that could serve the same purpose.

**Special Issue:**

No

---

> ### Author Response · Authors · 2024-03-14
> **Rebuttal for Reviewer dLtb (1/2)**
>
> The main concerns of the Reviewer revolve around the choices and justification on the deep learning framework. We slightly reorder the specific questions for clarity.
>
> &ensp; **1. Is there a justification on why SimCLR (2020) is used here? There are a plethora of recent self-supervised encoders and feature extractors that could serve the same purpose.**
>
> We thank the Reviewer for giving us the opportunity to explain. The self-supervised pretraining is done on the image we are analyzing, which is a rather small set to expect the model to learn relevant features training it from scratch. As we specify in the updated section 2.3. Feature extraction we use, to our knowledge, the only publicly available model trained on different histopathology images (Ciga *et al.*, 2022). Specifically, it is a ResNet18 trained on 57 datasets using SimCLR (Chen *et al.*, 2020) self-supervised contrastive learning. We chose to use the same self-supervised method (SimCLR) to fine-tune their model hoping that this would keep the general histopathological feature extraction power but would adapt it to our dataset. While it is true there are other self-supervised method that could have been used as encoders, we think this combination of using a model thoroughly pretrained and employing their same method to fine-tune it to our task was the most efficient way of getting meaningful features for the task.
>
> ---
>
> &ensp; **2. Have you thought about keeping the larger patch size (128), but extracting multiple hidden layers of features from the model (i.e. not only the last one)? That would maybe help obtain local (early layers) and global (last layer) patch features.**
>
> We thank the Reviewer for their suggestion. We did consider utilizing earlier layers and using larger patches for guiding the cell typing however, we had two concerns about this:
> * We update Appendix F. Effect of patch size to show that big patches for neighboring cells have large overlap and would look almost exactly the same and thus the extracted features would be too close in the feature space to be able to use them to guide the permutations.
> * We are limited to using ResNet18 as the encoding network (see comment 1), which is quite shallow compared to other architectures. The amount of convolutional operations the patches undergo until the last layer of a shallow network can be arguably equivalent to the “early layers” in deeper architectures. Thus, we think that early layers in a ResNet18 would be "too early" for the network to extract any valuable information. These observations are shared by previous studies (Guérin et al., 2021).
>
> ---
>
> &ensp; **3.  The method section on the deep learning application is not justified as well as the other section**
>
> Answering the Reviewer’s **Questions 1 and 2**, we realized some details from section 2.3. Feature extraction were missing and thus decided to re-write the section including them:
>
> > Our dataset was too small to expect the model to learn relevant features training it from scratch. We therefore started from a publicly available ResNet18 model trained by self-supervision with SimCLR \cite{chen2020simple}  on 57 histopathological datasets \cite{ciga2022self}. We fine-tuned the model using the detected nuclei as centers, extracted one image patch per cell, and trained by self-supervision with SimCLR in the same way as in the original model. Experiments included patch sizes of $32\times32$ (DL: 32), $64\times64$ (DL: 64) and $128\times128$ (DL: 128) to test the contribution of different contexts. For example patches, see Appendix F. Finally, we used the fine-tuned model's last fully connected layer before prediction to define features for each cell patch.
>
> We additionally expanded Appendix F. Effect of patch size by including the images of two neighboring cells to better address the effect of patch size:
>
> > Figure 7 shows the different patch sizes extracted with the same neighboring nuclei as centers. We hypothesize $32\times32$ patches do not include enough context for the self-supervised method to learn meaningful representations, while $128\times128$ patches include too much context and nuclei that are close start having similar features, hindering their separation per type.
>
> ---
>
> &ensp; **4. I would maybe suggest to rethink about the order of subsections in the paper. Usually, all dataset subsections (simulated, synthetics, real) would be combined under the same section and be presented before any method section.**
>
> While we agree with the Reviewer that this is not the traditional layout for a scientific paper, we hope the Reviewers agree that this presentation, showing the actual order of the experiments, allows the reader to better follow the paper.
>
> ---

---

> ### Author Response · Authors · 2024-03-14
> **Rebuttal for Reviewer dLtb (2/2)**
>
> **References**
>
> Chen, T., Kornblith, S., Norouzi, M., & Hinton, G. (2020, November). A simple framework for contrastive learning of visual representations. In International conference on machine learning (pp. 1597-1607). PMLR.
>
> Ciga, O., Xu, T., & Martel, A. L. (2022). Self supervised contrastive learning for digital histopathology. Machine Learning with Applications, 7, 100198.
>
> Guérin, J., Thiery, S., Nyiri, E., Gibaru, O., & Boots, B. (2021). Combining pretrained CNN feature extractors to enhance clustering of complex natural images. Neurocomputing, 423, 551-571.

---

### Official Review · Reviewer_gGXC · 2024-03-06

**Confidence:** 3
**Preliminary Rating:** 3
**Recommendation:** Poster
**Final Rating:** 4

**Summary:**

This paper proposes to use cell "morphology" features to guide cell type deconvolution, i.e., similar features should be assigned to same cell type. The authors proposed to search for permutation of assigned cell types (or cluster memberships) that maximizes some clustering quality criterion, i.e., Calinski-Harabasz score. The authors proposed to find permutation first locally at spot-level, then globally to reduce computation cost. The authors demonstrated their method is superior to baseline deconvolution method in correctly identifying cell types on synthetic and semi-synthetic dastasets.

**Strengths:**

- This paper argues that cell type deconvolution method should take into account of cell morphology. This is a novel insights that can prove useful in cases where each cell type do exhibit distinct morphology on H&E slides.
- The method is relatively simple to understand, though it's clarify could be improved if the authors can omit unnecessary details.
- This paper setup synthetic dataset that is essential to show their method works since ground truth cell type is not available in real datasets. This is very helpful.

**Weaknesses:**

- The clarity of the paper could be improved. In particuilar, key information is missing in the method section of the paper. Additionally, the paper would benefit from additional discussions. More details in further comments.
- Experiments that would support that the necessity of the hierarhical approach is missing. More details in further comments.

**Detailed Comments:**

+ explain/introduce tissue morphology as early as possible.
+ define dimension for P_m: currently it seems to be overloaded. Specifically P_m can be multiplied with both A (in Equation 1) and X_m (in Equation 3) that are of matrices of different sizes.
+ Section 2.1: The authors should provide more details on how the optimization problems in Equation (3,5) are solved. Do you do a search over all permutations and pick one that maximizes the objective?
+ Page 3: what is the "new space of acceptable permutations" ? Define this concretely.
+ Figure 1: label the different colors or at least mention what they represent.
+ Page 3 "Having significantly reduced the permutation space": Provide concrete numbers on  the reduction of the permutation space. The permutation space goes from L to something much smaller than L?
+ In Appendix A. Provide exact numbers for N, L, M, K etc. Readers have no idea what these numbers are and it's important that they are clearly written out.
+ The authors should have experiments that illustrate how much faster the hierarchical approach is compared to non-hierarchical approach, for varying values of N, L, K, M. The authors should have rough numbers on efficiency gain if N, L, K, M are set based on realistic dataset/scenarios.
+ Since the paper hinges upon assumptions that same cell type have similar morphology, the authors would want to discuss in more details on how accurate this assumption reflect real world use cases. It would be nice to provide examples that this assumption fails as well.

**Justification Of Final Rating:**

The authors addressed most of my comments, in particular why the hierarchical approach is necessary. It would still be nice if the authors could do experiments that compare runtime of global/hierarchical approaches. However, the justification that the global approach is too costly given is convincing.

I'll raise the rating to 4.

**Justification Of The Preliminary Rating:**

This paper solves an important problem in cell type deconvolution, the method is relatively simple and experiments on synthetic datasets are strong indicators that the method works. However, the paper lacks key information and its clarity could be improved significantly. Additionally, the authors would need to add additional experiments to support their contribution. Therefore, I recommend borderline.

**Questions To Address In The Rebuttal:**

Address the previous comments, e.g., make the paper more clear and add additional experiments to support the stated contribution.

**Special Issue:**

No

---

> ### Author Response · Authors · 2024-03-14
> **Rebuttal for Reviewer gGXC (1/2)**
>
> &ensp; **1. Explain/introduce tissue morphology as early as possible**
>
> Following the Reviewer's comment, in the updated Introduction we introduced *tissue morphology* in the very first sentence as being the *spatial organization of the tissue*:
>
> >Spatial transcriptomics has advanced our ability to understand the interplay between gene expression and tissue morphology i.e. spatial organization of tissue.
> ---
> &ensp; **2. Define dimension for $P_m$: currently it seems to be overloaded. Specifically $P_m$ can be multiplied with both A (in Equation 1) and $X_m$ (in Equation 3) that are of matrices of different sizes.**
>
> We thank the Reviewer for spotting this, this surely is an artefact of trying different notations to try and make the formulation as clear as possible. Following the Reviewer's comment, we added the dimensions to $P_m \in \\{ 0,1 \\} ^ {N \times N}$ and re-wrote $X \in \\{ 0,...,L \\} ^{N \times M}$, $P_m$ being the permutation matrix that permutes the positions of the cells inside a spot $m \in M$ and $X$ mirroring the dimensions of  $A \in \\{ 0,1 \\} ^ {N \times M}$ but indicating the $\\{ 1,...,L \\}$ cell identity labels.
>
> Now in **Equation 1**, $P_m$ multiplies $A$ and we impose the constraint on restricting the permutations inside a single spot. In **Equation 3**, $P_m$ multiplies $X_m$ which is the cell type matrix for a single spot $m \in M$ of dimension $X_m \in \\{ 1,...,L \\}^{N \times 1}$.
>
> ---
> &ensp; **3. Section 2.1: The authors should provide more details on how the optimization problems in Equation (3,5) are solved. Do you do a search over all permutations and pick one that maximizes the objective?**
>
> Yes, as the Reviewer indicates, for both we try all permutations which is a manageable task due to the constraints we impose locally and the reduction of the permutation space globally. Following the comment, we specify this by adding *exhaustively* to both Equations.
>
> ---
> &ensp; **4. Page 3: what is the "new space of acceptable permutations" ? Define this concretely.**
>
> We tried to define it in the previous paragraph but, following the Reviewer's comment, we will rewrite it to further clarify it:
>
> After the local arrangement per spot...
> >  However, it does not optimize arrangements within spots where the number of cells for each cell type is identical $ |X_{m,i}| = |X_{m,j}| \quad \textrm{for} \quad i \neq j $. In the case where $|X_{m,i}| = |X_{m,j}| = 1$, it is not possible to calculate the CH score, while if $|X_{m,i}| = |X_{m,j}| > 1$ there are multiple $P_m$ that yield the same highest CH score.
>
> > With these $m$ spots we define a new space of acceptable permutations $\mathcal{P'}_m \subset \mathcal{P}_m$.
>
> ---
> &ensp; **5. Figure 1: label the different colors or at least mention what they represent.**
>
> We agree with the Reviewer and added in the caption:
>
> > Colors represent different cell identities.
>
> ---
> &ensp; **6. Page 3 "Having significantly reduced the permutation space": Provide concrete numbers on the reduction of the permutation space. The permutation space goes from L to something much smaller than L?**
>
> This is related to **Question 8**, so we will answer them together. The answer to  **Question 6, 7 and 8**, led to the rewriting of Appendix A to add these concerns.
>
> The gain in speed and efficiency compared to having a non-hierarchical permutation scheme is, in most cases, infinite. If one were to directly perform the permutations without considering the constraints and the hierarchical scheme, it would not be possible to apply this to anything more than 9 cells, as the permutation operation has a $O(n!)$ complexity so 10 cells would require $10! = 3 628 800$ which, at least in a home computer, is a prohibitive number of operations.
>
> If one were to consider at least the constraints and try to solve globally, for all possible combinations of permutations within every spot, it would quickly become prohibitive as well. Consider even the simulated example in Figure 2. If one were to try to globally optimize it, without the hierarchical step, it would take (from left to right and from top to bottom): $1 \cdot 2 \cdot 20 \cdot 12 \cdot 5 \cdot 12 \cdot 10 \cdot 1 \cdot 1 \cdot 3 \cdot 6 \cdot 6 = 31 104 000$  operations which is prohibitive, while the hierarchical approach enables to reduce to space of acceptable permutations to $1 \cdot 2 \cdot 2 \cdot 2 \cdot 1 \cdot 2 \cdot 1 \cdot 1 \cdot 1 \cdot 1 \cdot 6 \cdot 2 = 192$. Specifically regarding varying values of N, L, K, M, in anything even close to a realistic scenario would directly mean an infinite gain in efficiency.
>
> The only concern left would be whether this hierarchical approach achieves as good results as directly a global optimization, and that is what we show in Figure 4, where we do  $10 000$ random global permutations per spot to see if our hierarchical approach gets close to the best arrangements.

---

> ### Author Response · Authors · 2024-03-14
> **Rebuttal for Reviewer gGXC (2/2)**
>
> &ensp; **7. In Appendix A. Provide exact numbers for N, L, M, K etc. Readers have no idea what these numbers are and it's important that they are clearly written out.**
>
> If the Reviewer refers to the specific numbers for the simulated data in Figure 2, we agree and we added the specific numbers in Appendix A:
>
> > Specifically, for the example in Figure 2 we used $M=12$, $N_m=[2,6]$, $L=4$ and $K=10$. We chose the proportion of cells $p_l$ such that the first row contains a higher percentage of cell type $C_3$, the second $C_1$ and the third $C_4$, with $C_2$ being only marginally present, simulating different layers.
>
> ---
>
> &ensp; **8. The authors should have experiments that illustrate how much faster the hierarchical approach is compared to non-hierarchical approach, for varying values of N, L, K, M. The authors should have rough numbers on efficiency gain if N, L, K, M are set based on realistic dataset/scenarios.**
>
> The Reviewer is suggesting additional experiments but we hope the theoretical explanations in **Question 6** will address their concerns. We added these explanations to Appendix A for further clarity.
>
> ---
>
> &ensp; **9. Since the paper hinges upon assumptions that same cell type have similar morphology, the authors would want to discuss in more details on how accurate this assumption reflect real world use cases. It would be nice to provide examples that this assumption fails as well.**
>
> This is a very case-dependent question and should always be confirmed by clinical collaborators on a case to case basis. But there is one general consideration is that whichever cell types have an identifiable morphology in the stain at use (in this case H&E), are good candidates. For instance, in the specific case we are analyzing and sharing in the real data application (Tangram applied to mouse brain cortex), oligodendrocytes usually have a small and round nuclei surrounded by cytoplasm, which gives it a distinct morphology -and thus morphological features-. Conversely, trying to pinpoint the morphology of cells in different neuronal layers from the itra-encephalic (IT) or the pyramidal tract (PT) can be challenging in H&E as their differences may not be visible in the cell nuclei or the extracellular matrix, which is what H&E stains for.
>
> Following the Reviewer's comment, we added this to the Discussion of the paper:
>
> > An inherent limitation of the method lies in the assumption that every cell type possesses an identifiable morphology that can be leveraged in this problem, which may not be applicable for every encountered cell type. For instance, non-neuronal cells like oligodendrocytes have an identifiable small and round nucleus surrounded by cytoplasm, while cells in different neuronal layers can be more challenging to distinguish in H\&E.

---

### Author Response · Authors · 2024-03-14
**To all Reviewers**

We sincerely thank the Reviewers for their feedback. Specifically, we are grateful to **Reviewer gGXC** and **Reviewer W2yE** for acknowledging the novelty and usefulness of the work, and to **Reviewer dLtb** and **Reviewer W2yE** for recognizing the availability of the source code. **Reviewer gGXC**'s positive remarks on the synthetic experiments and **Reviewer dLtb**'s remarks on the clarity of the paper and additional appendices were really encouraging. Further, we are glad that **Reviewer W2yE** highlighted the contribution of our method to the field of cell type deconvolution in spatial transcriptomics.

We appreciate **Reviewer gGXC**,  **Reviewer dLtb** and **Reviewer W2yE** for providing us with the opportunity to clarify the methodology, deep learning framework and applicability, respectively. We hope our answers addressed all questions.

---

### Meta-Review · Area_Chair_U8Eo · 2024-04-03

**Recommendation:** Accept (Poster)
**Confidence:** 5

**Metareview:**

All reviewers have recommended a weak accept for this manuscript following the rebuttal. The meta-reviewer is pleased to recommend its acceptance, upon the reviewers' commitment to thoroughly incorporate all reviewers' questions and comments in the final revised paper.

---

### Decision · Program_Chairs · 2024-04-05

Accept (Poster)